# Psychological Interventions and Bariatric Surgery among People with Clinically Severe Obesity—A Systematic Review with Bayesian Meta-Analysis

**DOI:** 10.3390/nu14081592

**Published:** 2022-04-12

**Authors:** Dawid Storman, Mateusz Jan Świerz, Monika Storman, Katarzyna Weronika Jasińska, Paweł Jemioło, Małgorzata Maria Bała

**Affiliations:** 1Chair of Epidemiology and Preventive Medicine, Department of Hygiene and Dietetics, Jagiellonian University Medical College, 31-034 Krakow, Poland; dawid.storman@doctoral.uj.edu.pl (D.S.); mateusz.swierz@doctoral.uj.edu.pl (M.J.Ś.); 2Systematic Reviews Unit, Jagiellonian University Medical College, 31-034 Krakow, Poland; monika.storman@wp.pl; 3Department of Diabetology and Internal Medicine, Medical University of Warsaw, 02-091 Warszawa, Poland; 4Students’ Scientific Research Group, Systematic Reviews Unit, Jagiellonian University Medical College, 31-034 Krakow, Poland; katarzyna.jasinska100@gmail.com; 5Faculty of Electrical Engineering, Automatics, Computer Science and Biomedical Engineering, AGH University of Science and Technology, al. Mickiewicza 30, 30-059 Krakow, Poland; pawljmlo@agh.edu.pl

**Keywords:** bariatric surgery, psychological interventions, weight loss, cognitive-behavioural therapy, systematic review

## Abstract

Aim: To assess the effectiveness of perioperative psychological interventions provided to patients with clinically severe obesity undergoing bariatric surgery regarding weight loss, BMI, quality of life, and psychosocial health using the Bayesian approach. Methods: We considered randomised trials that assessed the beneficial and harmful effects of perioperative psychological interventions in people with clinically severe obesity undergoing bariatric surgery. We searched four data sources from inception to 3 October 2021. The authors independently selected studies for inclusion, extracted data, and assessed the risk of bias. We conducted a meta-analysis using a Bayesian approach. PROSPERO: CRD42017077724. Results: Of 13,355 identified records, we included nine studies (published in 27 papers with 1060 participants (365 males; 693 females, 2 people with missing data)). Perioperative psychological interventions may provide little or no benefit for BMI (the last reported follow-up: MD [95% credible intervals] = −0.58 [−1.32, 0.15]; BF_01_ = 0.65; 7 studies; very low certainty of evidence) and weight loss (the last reported follow-up: MD = −0.50 [−2.21, 0.77]; BF_01_ = 1.24, 9 studies, very low certainty of evidence). Regarding psychosocial outcomes, the direction of the effect was mainly inconsistent, and the certainty of the evidence was low to very low. Conclusions: Evidence is anecdotal according to Bayesian factors and uncertain whether perioperative psychological interventions may affect weight-related and psychosocial outcomes in people with clinically severe obesity undergoing bariatric surgery. As the results are ambiguous, we suggest conducting more high-quality studies in the field to estimate the true effect, its direction, and improve confidence in the body of evidence.

## 1. Introduction

Recently, the prevalence of obesity has dramatically increased worldwide and has been estimated at 600 million people worldwide [1,2,3]. Nowadays, this problem affects not only high-income countries but low- and middle-income countries as well [4].

Obesity is a chronic condition and is classified in chapter E66 of the International Classification of Diseases [5]. It is characterized by excessive accumulation of adipose tissue, and according to the most common classification (World Health Organisation) is recognized when Body Mass Index (BMI) is ≥30 kg/m^2^ in adults [6].

Clinically severe obesity (CSO) is defined as a BMI of at least 40 kg/m^2^ or at least 35 kg/m^2^ with comorbid conditions such as type 2 diabetes, hypertension, dyslipidaemia, obstructive sleep apnoea, or stress urinary incontinence. People with CSO are more frequently affected by psychosomatic disorders and premature death compared to people without CSO because the condition influences disability, psychosocial well-being, and quality of life (QoL) [7,8,9].

Currently, bariatric surgery (BS) is the most effective treatment people with CSO [10,11,12]. However, it can be associated with adverse postsurgical outcomes, i.e., weight regain, occurrence of maladaptive eating behaviours, deterioration of the QoL, and others [13,14,15,16].

International guidelines and recommendations highlight a need for employing a multidisciplinary approach of care incorporating different types of support: psychological, dietary, or physical activity [17,18,19]. However, there are no recommendations on the type of perioperative psychological interventions (PPIs) and their optimal timing with respect to surgery [20]. These interventions could be helpful for patients who, due to low self-confidence, low self-efficacy, rigid patterns of behaviour, or cognitive schemas, find it difficult to comply with postoperative restrictions [21,22,23,24,25].

There is evidence showing the positive effect of behavioural interventions alone on weight loss (WL) in people with obesity and assessing their impact on improvement of comorbidities [26,27,28]. PPIs may play a significant role [29]; on the basis of cognitive-behavioural approach, these help to modify antecedent, behaviour, consequences, and thoughts, taking into account emotions, relationship, mindfulness, acceptance, values, goals, and meta-cognition, which are believed to maintain a positive energy balance [30,31,32,33]. According to Kulick et al. [34], PPIs seem to be beneficial as they provide different perspectives, coordinated expertise and skills, and sufficient patient engagement. Moreover, a greater focus on psychosocial functioning may also optimize post-operative weight outcomes [21,22,25,35,36]. According to Brennan et al. [37], the mechanism of action of PPIs for people with obesity is still unclear. So far, several systematic reviews have been published assessing the efficacy of perioperative interventions. However, several of them did not attempt to quantitatively summarise the data with meta-analysis [38,39,40,41,42] or they did not refer to the registered protocol [43,44,45], focused primarily on weight-related outcomes [20,46], did not take into account the outcomes we were interested in [47], or addressed a different population [48]. Thus far, there are no systematic reviews conducted according to the state-of-the-art methodological standards [49] in the discussed area.

The objective of this study was to assess the effectiveness of PPIs provided during the perioperative period to patients with CSO undergoing BS regarding WL, QoL, and psychosocial health.

## 2. Materials and Methods

We included randomized controlled trials (RCTs) that had to follow participants for a minimum of six months (time frame refers to the intervention itself or a combination of the intervention with a follow-up phase). This study was reported according to the Preferred Reporting Items for Systematic Reviews and Meta-Analyses (PRISMA) statement [50]. The PRISMA checklist is included in the Appendix A. A protocol has been registered in PROSPERO: CRD42017077724.

### 2.1. Eligibility Criteria

#### 2.1.1. Population

We included studies on people of any age with CSO during the perioperative period (after qualification for BS or up to any time post-surgery).

#### 2.1.2. Intervention

We defined PPIs as interventions aimed at changing habits, diet, or physical activity through cognitive (together with psychoeducation) and/or behavioural strategies [45,51,52]. They had to be provided in the form of structured interactions between participants and facilitators (psychologists, psychotherapists, therapists in training, or other trained professionals supervised by a clinical psychologist or therapist) [53]. We did not include interventions focusing solely on physical activity, social support, or dietary advice. Due to the lack of a strict definition of the perioperative period, we planned not to limit the starting timepoint of PPI.

#### 2.1.3. Outcomes

All outcomes had to be measured at baseline (before the start of PPI) and at least six months post surgery, using a validated tool. Primary outcomes were: change in BMI, WL (kg, %WL), change in self-efficacy, and change in QoL. Secondary outcomes were: assessment of maladaptive eating behaviours (such as binge eating, grazing, or emotional overeating) [54], change in psychological symptoms (anxiety, depression), change in problems with relationships, change in cognitive function (memory improvement, executive function, attention), change in alcohol and other substances misuse, and change in suicidal behaviour.

### 2.2. Search Strategy

We searched the following electronic databases from inception to 3 October 2021: MEDLINE Ovid, Embase, Cochrane Central Register of Controlled Trials (CENTRAL), and ClinicalTrials.gov (search strategies are enclosed in Appendix A) without any language restrictions. We checked references of included studies for additional studies.

### 2.3. Study Selection

Pairs of authors (D.S., P.J., M.J.S., M.S. and K.W.J.) independently screened titles and abstracts and then full texts against eligibility criteria using Covidence software^®^ and Rayyan QCRI. Any disagreement was resolved through discussion or consultation with another reviewer (MMB).

### 2.4. Data Extraction

Pairs of reviewers (D.S., P.J., M.J.S., M.S. and K.W.J.) independently extracted data on study settings, time frame, methods, details of population, intervention, and outcomes. We resolved any discrepancies by discussion. One review author (D.S.) additionally checked all extracted data again.

Attempts were made to contact corresponding authors in case of missing data or when any clarification was required.

### 2.5. Risk of Bias (ROB) Assessment

Pairs of reviewers (D.S., P.J., M.J.S., M.S. and K.W.J.) independently assessed ROB in every study using the Cochrane risk of bias assessment tool, according to the Cochrane Handbook [49,55]. Any disagreement was resolved by discussion or consultation with another reviewer (MMB).

### 2.6. Data Analysis and Synthesis

We expressed continuous data as mean differences with 95% credible intervals (CrI). We calculated pooled estimates using the random-effects model, as we believed there would inevitably be heterogeneity among the included trials [56]. The outcomes were estimated using Bayesian normal priors in JASP [57]. We provided Bayes factors (BF_01_) and used Lee and Wagenmakers’ thresholds for interpretations [58]. We used Markov Chain Monte Carlo sampler with four chains. Heterogeneity was assessed by analysing τ (group-level standard deviation). It was determined non-significant when ≤1 [59,60]. If performing meta-analysis was not possible, we presented results descriptively.

### 2.7. Assessment of Reporting Bias

For the investigation of small-study effect that could possibly explain publication bias, at least 10 studies should be included for a certain outcome to be able to produce a viable funnel plot or to run statistical tests for interpretation [61]. As we identified few studies for inclusion, we could not produce funnel plots for our comparisons.

### 2.8. Certainty of Evidence

We presented the overall certainty of evidence and justifications for each outcome separately as a “Summary of findings” table, in accordance with the GRADE approach. Two review authors (D.S., K.W.J.) independently rated the certainty of the evidence for each outcome.

### 2.9. Sensitivity Analysis

We performed a sensitivity analysis comparing studies with high, unclear, and low ROB for incomplete outcome data and selective outcome reporting. Additionally, we decided to analyse the PPI effect in subgroups according to the procedures, i.e., restrictive (e.g., LAGB, VBG, or SG), mixed (e.g., RYGB or GB), or both (post hoc analysis).

## 3. Results

We identified nine trials (published in 27 papers) out of 13,355 references, and 16 were labelled as ongoing (Appendix A). We presented details of the study flow on a PRISMA flow diagram (Figure 1) [56]. For detailed characteristics of included studies, see Table 1 and Appendix A.

### 3.1. Included Studies

#### 3.1.1. Participants

Overall, 1060 participants were included. In total, 533 were randomized to intervention and 527 to a comparator group (no information was provided about nine participants). The total sample size ranged from 36 [68] to 240 [62].

All trials reported mean BMI at baseline, which ranged from 35.4 kg/m^2^ to 51.4 kg/m^2^ (mean 45.2 ± 6.6) in the intervention group and from 36.5 kg/m^2^ to 50.90 kg/m^2^ (mean 44.5 ± 6) in the control group. The proportion of females ranged from 60.3% to 90.1% (mean 75.2%) in the intervention group and from 67% to 94.4% (mean 80%) in the control group. Mean age ranged from 43.5 to 51.0 years (mean 44.7 ± 9.9) in the intervention group, and from 39.3 to 53.9 years (mean 44.2 ± 10.3) in the control group (one study reported mean age of the whole sample to be 40.18 years [66]). Participants underwent different types of BS: Roux-en-Y-gastric by-pass (six studies), laparoscopic adjustable gastric banding (three studies), vertical banded gastroplasty (two studies), sleeve gastrectomy (one study), unspecified gastric bypass (two studies), laparoscopic sleeve gastrectomy (one study), and laparoscopic Roux-en-Y-gastric by-pass (one study).

#### 3.1.2. Intervention

Two studies provided PPIs only before BS [63,70], four trials only after BS [66,67,68,69], and three both before and after surgery [62,64,65]. Three studies applied intervention in the form of group sessions, four individually, and two as both individual and group sessions. The mean number of sessions was 10 (from 3 to 27). Duration of intervention ranged from six weeks [69] to over two years [64], and the longest follow-up was 4 years and 4 weeks [63]. Single session duration ranged from 15 to 180 min. Two studies included interventions that were multidisciplinary [64,65], five trials focused on behavioural-related approaches such as cognitive-behavioural therapy and behavioural therapy [62,63,66,68,70], and two trials focused solely on education-related interventions [67,69].

#### 3.1.3. Outcomes

The provided primary outcomes were: change in BMI, data on WL (kg or %), change in self-efficacy, change in QoL.

Among reported secondary outcomes were the occurrence of maladaptive eating behaviours and change in the severity of psychological symptoms. None of the research provided information about the change in problems with relationships, in cognitive function, in suicidal behaviour, or in alcohol and other substance misuse.

### 3.2. Excluded Studies

A list of excluded studies with reasons is provided in Appendix A.

### 3.3. ROB in Included Studies

Detailed ROB assessment is presented in Appendix A. For an overview of reviewers’ judgments on each ROB item for individual studies and across all research, see Appendix A. We assessed two trials to be at high risk of bias on four domains [62,64], two studies on three domains [63,65], three studies on two domains [67,69,70], and two trials on one domain [66,68].

### 3.4. Effects of Interventions

See summary of findings table (Table 2) for the main comparison “PPI in patients with CSO undergoing BS”.

#### 3.4.1. Primary Outcomes

Each of the weight-related outcomes were analysed at three timepoints (6–12 months follow-up, 1–2 years follow-up, and the last reported follow-up of the study).

##### Weight-Related Outcomes

Pooling the studies in a random-effects meta-analysis demonstrated no differences between the intervention and control groups in BMI change from baseline at any of the investigated follow-ups: 6–12 months (−0.29 [−1.6, 0.83]; BF_01_ = 1.54), 1–2 year (−0.59 [−1.34, 0.12], BF_01_ = 0.59), and the last follow-up (−0.58 [−1.32, 0.15], BF_01_ = 0.65) (Figure 2, Appendix A, Appendix A).

Upon pooling, we did not observe any differences between intervention and control groups in WL at any investigated follow-up (0.14 [−1.43, 1.99]; BF_01_ = 0.44, −0.56 [−2.2, 0.66]; BF_01_ = 1.18, −0.50 [−2.21, 0.77]; BF_01_ = 0.24 for 6–12 months, 1–2 year, and the last follow-up, respectively) (Figure 3, Appendix A).

Upon performing meta-analysis, we did not observe any differences in percentage WL between the intervention and control groups at any follow-up time-point (−1.60 [−4.68, 1.48], −0.54 [−2.79, 1.07]; BF_01_ = 1.17, −1.06 [−4.53, 0.92]; BF_01_ = 0.88, for 6–12 months, 1–2 year, and the last follow-up, respectively).

##### Psychosocial Outcomes

Because of heterogeneity in the outcomes’ presentation, we could not summarize data quantitatively thus only descriptive analysis is presented.

Change in self-efficacy was provided in one study. It was measured using the General Self Efficacy Scale. The direction of the effect was consistently in favour of intervention in the 6–12 months and the last reported follow-ups and favour of control in the 1–2 years’ follow-up (Appendix A). The certainty of evidence was very low.

Change in QoL was provided in three studies. In the included trials, several questionnaires were used: 36-Item Short Form Survey—SF-36 (mental and physical components), Schedule for the Evaluation of Individual Quality of Life—SEIQoL, and Impact of Weight on Quality of Life—IWQOL. The direction of the effect was inconsistent (Appendix A). The certainty of evidence was very low for every follow-up.

#### 3.4.2. Secondary Outcomes

The trials described different eating behaviours within binge eating or episodes of bulimia. The direction of the effect was inconsistent (Appendix A). The certainty of evidence was very low in the 6–12-month and 1–2-year follow-ups and low for the last reported follow-up.

Two kinds of mood conditions were analysed in included studies: depressive and anxious. The utilised questionnaires included PHQ-9—Patient Health Questionnaire, HADS—Hospital Anxiety and Depression Scale, BDI—Beck Depression Inventory score, or POMS—Profile of Mood States. The direction of the effect was consistently in favour of intervention at 1–2 years’ follow-up. For the other follow-ups, it was inconsistent (Appendix A). The certainty of evidence was low for all the follow-ups.

#### 3.4.3. Sensitivity Analysis

We did not observe differences between the subgroups analysed in terms of risk of bias. Additionally, there were two subgroups regarding the type of procedure: mixed and mixed/restrictive but without statistical significant differences between the subgroups (Appendix A).

### 3.5. Certainty of the Evidence

We rated the certainty of evidence for the outcomes of interest as low or very low (Table 2).

## 4. Discussion

### 4.1. Summary of Main Results

This systematic review summarises eight RCTs examining the effect of PPI applied to people with CSO undergoing BS. Studies reported the effects of interventions with respect to BMI, WL, QoL, maladaptive eating behaviours, change in psychological symptoms, and self-efficacy, however inconsistently. Majority of trials did not report adequate information to assess the ROB, and six studies were assessed as high risk of bias on at least two domains.

There was heterogeneity in measurement tools, assessed outcomes, and unit of outcome measures. Based on the identified data, we could not demonstrate the benefit of PPIs with respect to BMI, WL, QoL, maladaptive eating behaviours, change in psychological symptoms, and self-efficacy. According to Bayes factors, efficacy of PPI on WL and BMI is based at most on anecdotal evidence. We found no data for problems with relationships, cognitive function, suicidal behaviour, or alcohol and other substances’ misuse.

### 4.2. Agreements and Disagreements with Other Studies or Reviews

In contrast to our results, there were several reviews where statistically significant reductions in post-operative weight and BMI were observed. In some of them, both RCTs and non-RCTs were included [45,46,48,71,72]. The effect of intervention presented in reviews where non-RCTs were considered should be analysed consciously because this study design is related to a higher RoB and the potential influence of confounding factors. David et al. noted significant benefits of psychosocial interventions for WL, but they did not synthesise the results [45]. In a study conducted by Świerz et al. [47], who defined perioperative period being 30 days before to 30 days after surgery, it was suggested that psychotherapy might have no effect on WL (the authors assessed the certainty of evidence as low). Stewart et al. [20], who considered any behavioural interventions, with the explicit aim of changing behaviour related to diet and/or physical activity, also found no significant difference in BMI change at 6- and 12-month follow-up after surgery, but regarding WL and percent excess weight change, they observed a greater change in the intervention groups after 2 years post-BS.

Among studies where the authors attempted to synthesize the psychological outcomes, Marshall et al. [48] demonstrated improvement in symptoms of depression, anxiety, and QoL after interventions provided by multidisciplinary team.

Bayesian methods are increasingly used in health care research, including meta-analyses. Nevertheless, to the best of our knowledge, this is the first systematic review in the field of PPI in BS, which utilises this methodology. The use of this approach allows accurate inference results despite the small sample size [73]. Compared to the classical one, it allows for more convenient determining which hypotheses (null or alternative) is more supported by the data (thresholds for Bayesian factors) [58,74].

Another significant strength of our study is including trials of any form of PPI. It also includes the effect of these interventions on other than weight-related outcomes, such as psychosocial outcomes (QoL, self-efficacy, maladaptive eating behaviours, etc.).

## 5. Conclusions

We explored the impact of different types of PPIs applied in people with CSO undergoing BS. Low to very-low certainty evidence suggests that PPIs might reduce weight regarding middle- and long-term post-surgery WL (follow-ups from 12 mo) and BMI. Additionally, the evidence is insufficient to conclude how PPI, and which specific type of PPI, affects psychosocial outcomes. We found no data for problems with relationships, cognitive function, suicidal behaviour, or alcohol, and other substances misuse.

Our results indicate no strong evidence basis to support the PPIs in people with CSO undergoing BS.

The results of this review should be interpreted with caution, as most of the evidence was rated as very low and low quality due to inconsistency, indirectness, or RoB for many of the outcomes measured.

High-quality trials with long-term follow-up are required to strengthen the body of evidence, as the current evidence is of low to very low methodological quality, and, at most, of anecdotal strength (Bayes factors). Furthermore, consistent measures of psychosocial outcomes using validated tools should be used in future research, and reports should provide adequate and transparent methodological details such as allocation concealment, blinding, attrition, selective reporting, and validity of tools.

## 6. Differences between the Protocol and Final Review

We added the analyses of the effect of PPIs in subgroups according to the procedures, i.e., restrictive (e.g., LAGB, VBG, or SG), mixed (e.g., RYGB or GB), or both (post-hoc analysis).

## Figures and Tables

**Figure 1 nutrients-14-01592-f001:**
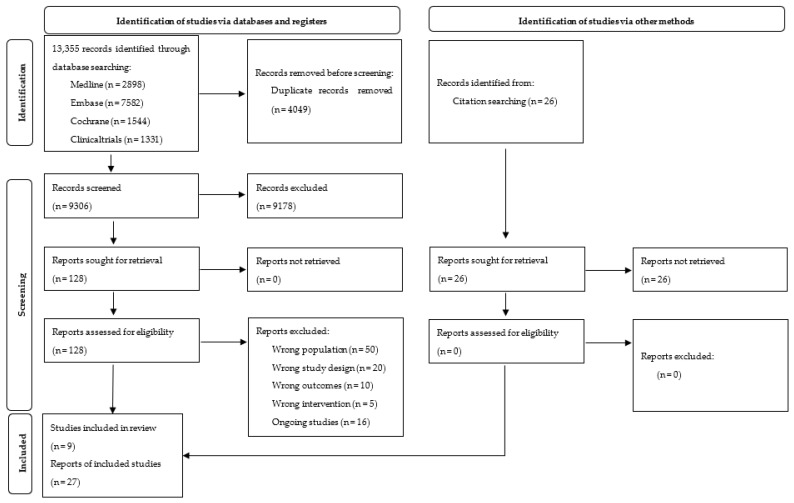
PRISMA 2020 Flow Diagram.

**Figure 2 nutrients-14-01592-f002:**
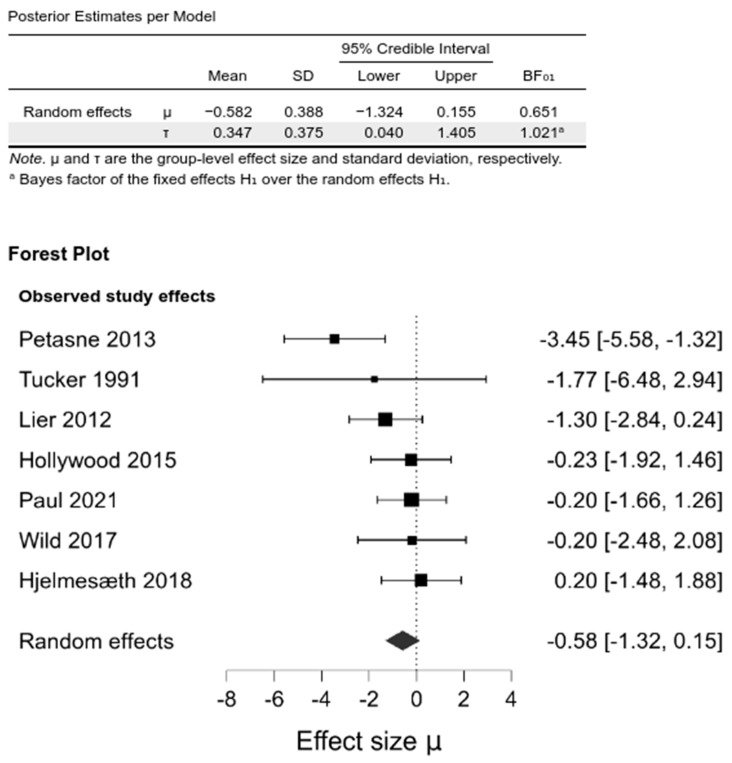
Comparison between psychological intervention versus any control in the change of BMI at the last follow-up.

**Figure 3 nutrients-14-01592-f003:**
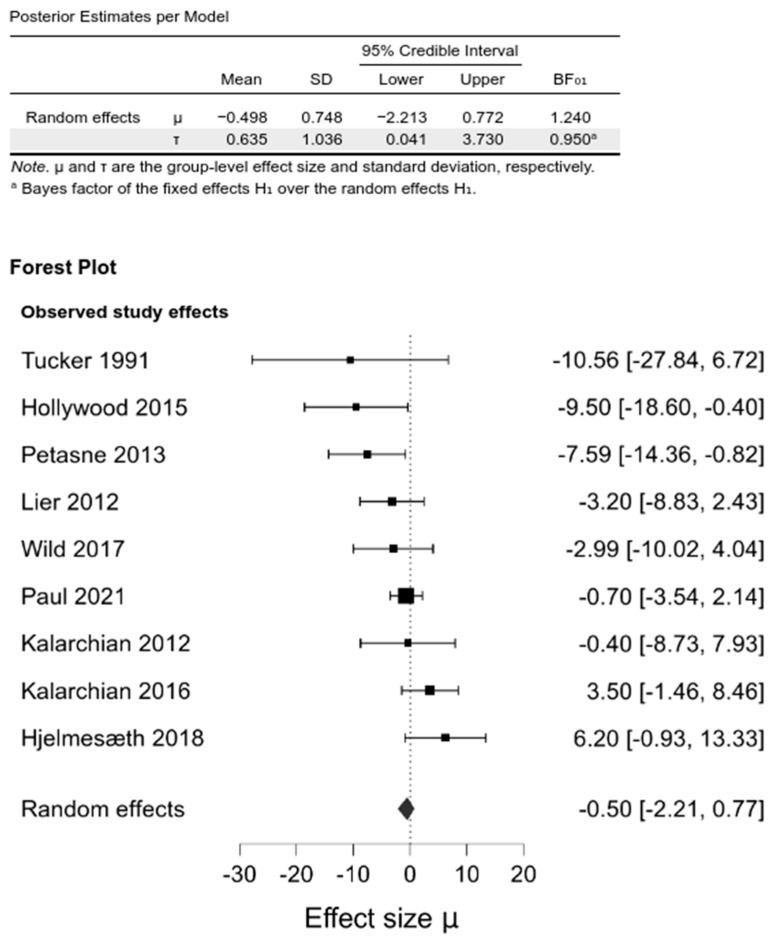
Comparison between psychological intervention versus any control in the WL at the last follow-up.

**Table 1 nutrients-14-01592-t001:** Characteristics of included studies.

Study Name (Country)	Intervention				Control				Type of Surgery	Time Frame of PPI ^#^	Follow-Up Post-Surgery (Months)	Outcomes	COI	Funding Reported
	Description	Randomized n	Age Years Mean (SD)	Female n (%)	Description	Randomized n	Age Years Mean (SD)	Female n (%)						
Kalarchian 2016 (USA) [62]	6-month manualized behavioural lifestyle intervention: diet (1200–1400 calories/day) and PA goals + 12 individual, 1 h face-to-face counselling sessions pre-surgery + 12 telephone calls (15–20 min) pre-surgery + 3 monthly contacts + CAU	121	43.9 (10.3)	64 (90.1%)	Synopsis of information provided in intervention group; CAU: presurgical physician supervised diet + activity program	119	45.9 (11.6)	65 (90.3%)	RYGB, LAGB	24 w before BS and 24 mo after BS	6, 12, 24	1,2,5,6	No	Yes
Hjelmesæth 2019 (Norway) [63]	10 weekly individual sessions before BS aiming to improve dysfunctional eating behaviours	50	44.1 (9.8)	27 (64.3%)	10 weeks of nutritional support and education pre-surgery	52	41.2 (9.6)	28 (73.7%)	RYGB, SG	12 w before BS	12, 48	1,2,5,6	No	Yes
Lier 2012 (Norway) [64]	CBT (1 preoperative group session/week for 6 weeks + 3 postoperative group sessions (6 months, 1 year and 2 years post-surgery)—Semi-structured therapy manual	49	43.5 (11.1)	36 (74.0%)	CAU: 1 pre- and 1 post- surgery 4 h educational seminar on dietary strategies and behaviours	50	42.4 (9.1)	32 (67.0%)	GB	6 w before BS and 24 mo after BS	12	2	No	Yes
Hollywood 2015 (UK) [65]	Bariatric rehabilitation service: 3 one-to-one 50 min sessions with psychologist 2 weeks pre-surgery, before discharge and at 3 months post-surgery + CAU	82	45.6 (11.1)	61 (74.4%)	CAU: Standard diet sheet postoperatively	80	44.8 (10.6)	61 (76.2%)	RYGB	2 w before BS and 3 mo after BS	12	1,2,4,5,6	NR	Yes
Tucker 1991 (USA) [66]	Eating- and lifestyle-related materials every 2 weeks for 24 weeks post-surgery + 6 monthly consultations + CAU	41 ^†^	40.18 ^†^	21 (65.6) ^†^	CAU: Basic pre-surgery info on necessary eating-behaviour changes	41 ^†^	40.18 ^†^	21 (65.6) ^†^	GB, VBG	6 m after BS	6, 12, 24	1,2	NR	NR
Wild 2015 (Germany) [67]	CAU + 1 year supervised video- conferencing-based psychoeducational group: eight 90 min face-to-face and six 50 min videoconferencing sessions + education in nutrition and exercise	59	41.2 (9.0)	35 (60.3%)	CAU: Conventional surgical visits at 1, 3, 6, and 12 months post-surgery	58	41.9 (9.6)	45 (80.4%)	LSG, RYGB, LAGB	12 m after BS	6, 12, 37.9	1,2,3,4,5,6	No	Yes
Kalarchian 2012 (USA) [68]	6-month behavioural intervention (6.6 year after surgery): instruction to intake 1200–1400 calorie/day and to follow postoperative guidelines + exercise program + 1 h face-to-face group meetings (12 weekly meetings) + 15–20 min telephone coaching (5 biweekly)	18	51.0 (7.6)	15 (83.3%)	Wait list control group	18	53.9 (6.6)	17 (94.4%)	GB, LAGB, VGB	79 m after BS	85, 91	2	No	Yes
Nijamkin 2013 (USA) [69]	6 nutrition and lifestyle education and behavioural–motivational group sessions every other week starting at 7 months post-surgery (use of Dietary Guidelines, nutrition education, exercises, motivational strategies) + e-mail reminders + telephone calls + CAU	72	44.2 (12.6)	62 (86.1%)	Brief printed guidelines; CAU: Optional counselling	72	44.8 (14.4)	58 (80.6%)	LRYGB	6 m after BS	12	1,2,6	NR	Yes
Paul 2021 (The Netherlands) [70]	cognitive behavioural therapy of 10 individual sessions of 45 min, conducted by a psychologist or cognitive behavioural therapeutic worker	65	44.1 (8.2)	46 (73%)	Conventional preparation procedure consisting of an information meeting by the surgeon or nurse practitioner and an information meeting by the dietitian. Patients also receive a detailed patient information booklet.	65	39.3 (10.6)	49 (75%)	GB	10 w before BS	12	1,2,4,5,6	No	NR

CAU—care as usual, COI—conflict of interest, GB—Gastric Bypass, SG—Sleeve Gastrectomy, RYGB—Roux-En-Y Gastric Bypass, CBT—Cognitive-behavioural therapy, BS—bariatric surgery, LAGB—Laparoscopic adjustable gastric banding, VGB—Vertical Banded Gastroplasty, LSG—laparoscopic sleeve gastrectomy, LRYGB Laparoscopic Roux-En-Y Gastric Bypass, NR—not reported, w—weeks, mo—months. ^†^ total (intervention and control groups); ^#^ includes the first time of PPI before BS and/or the last time of PPI after BS, if applicable. 1—Changes in measured BMI; 2—Weight loss; 3—Change in self-efficacy; 4—Change in quality of life; 5—Assessment of maladaptive eating behaviours; 6—Change in psychological symptoms.

**Table 2 nutrients-14-01592-t002:** Summary of findings table.

Psychological interventions in patients with morbid obesity undergoing bariatric surgery					
Patient or population: patients with obesity undergoing bariatric surgery					
Settings: any					
Intervention: any psychological interventions (such as BT/CBT/related to those, combined psychological intervention, education)					
Comparison: any control (such as care as usual care or minimal intervention, diet with physical activity, nutrition counselling)					
Outcomes	Control	Psychological Intervention	No of Participants (Studies)	Quality of the Evidence (GRADE)	Comments
Changes in BMI [kg/m^2^] Follow-up: 6 to 12 months	The mean change in BMI ranged across control groups from: −10.59 kg/m^2^ to −13.03 kg/m^2^	The mean change in BMI in the intervention groups was 0.29 kg/m^2^ lower (1.6 lower to 0.83 higher)	176 (2)	⊕ very low ^1^	Lower units indicate greater WL
Changes in BMI [kg/m^2^] Follow-up: 1–2 years	The mean change in BMI ranged across control groups from: −16.65 kg/m^2^ to −13.03 kg/m^2^	The mean change in BMI in the intervention groups was 0.59 kg/m^2^ lower (1.34 lower to 0.12 higher)	742 (7)	⊕ very low ^2^	Lower units indicate greater WL
Changes in BMI [kg/m^2^] Last follow-up	The mean change in BMI ranged across control groups from: −16.65 kg/m^2^ to −11.8 kg/m^2^	The mean change in BMI in the intervention groups was 0.58 kg/m^2^ lower (1.32 lower to 0.15 higher)	677 (7)	⊕ very low ^3^	Lower units indicate greater WL
WL [kg] Follow-up: 6 to 12 months	The mean WL [kg] ranged across control groups from: −37.90 kg to −29.75 kg	The mean WL [kg] in the intervention groups was 0.14 kg lower (1.43 lower to 1.97 higher)	416 (4)	⊕⊕ Low ^4^	Lower units indicate greater WL
WL [kg] Follow-up: 1–2 years	The mean WL [kg] ranged across control groups from: −46.18 kg to −30.7 kg	The mean WL [kg] in the intervention groups was 0.56 kg higher (2.20 lower to 0.66 higher)	842 (8)	⊕ very low ^5^	Lower units indicate greater WL
WL [kg] Last follow-up	The mean WL [kg] ranged across control groups from: −46.18 kg to −29.4 kg	The mean WL [kg] in the intervention groups was 0.50 kg higher (2.21 lower to 0.77 higher)	731 (9)	⊕ very low ^6^	Lower units indicate greater WL
WL [%] Follow-up: 6 to 12 months	See comment	See comment	143 (1)	⊕ very low ^7^	Higher units indicate greater WL Only one trial reported WL [%] within follow-up 6–12 months
WL [%] Follow-up: 1–2 years	The mean WL [%] ranged across control groups from: 29.4% to 30.1%	The mean WL [%] in the intervention groups was 0.54% lower (2.79 lower to 1.07 higher)	223 (2)	⊕⊕ Low ^8^	Higher units indicate greater WL
WL [%] Last follow-up	The mean WL [%] ranged across control groups from: 27.9% to 29.5%	The mean WL [%] in the intervention groups was 1.06% lower (4.53 lower to 0.92 higher)	204 (2)	⊕⊕ Low ^8^	Higher units indicate greater WL
Self-efficacy Follow-up: 6 to 12 months	See comment	See comment	97 (1)	⊕ very low ^9^	The direction of the effect was consistently in favour of intervention
Self-efficacy Follow-up: 1–2 years	See comment	See comment	110 (1)	⊕ very low ^9^	The direction of the effect was consistently in favour of control
Self-efficacy Last follow-up	See comment	See comment	74 (1)	⊕ very low ^9^	The direction of the effect was consistently in favour of intervention
Quality of life Follow-up: 6 to 12 months	See comment	See comment	115 (1)	⊕ very low ^9^	The direction of the effect was inconsistent
Quality of life Follow-up: 1–2 years	See comment	See comment	288 (3)	⊕ very low ^9^	The direction of the effect was inconsistent
Quality of life Last follow-up	See comment	See comment	251 (3)	⊕ very low ^9^	The direction of the effect was inconsistent
Maladaptive eating behaviours Follow-up: 6 to 12 months	See comment	See comment	205 (2)	⊕ very low ^9^	The direction of the effect was inconsistent
Maladaptive eating behaviours Follow-up: 1–2 years	See comment	See comment	366 (3)	⊕ very low ^9^	The direction of the effect was inconsistent
Maladaptive eating behaviours Last follow-up	See comment	See comment	498 (4)	⊕⊕ Low ^10^	The direction of the effect was inconsistent
Change in psychological symptoms Follow-up: 6 to 12 months	See comment	See comment	428 (3)	⊕⊕ Low ^10^	The direction of the effect was inconsistent
Change in psychological symptoms Follow-up: 1–2 years	See comment	See comment	498 (5)	⊕⊕ Low ^10^	The direction of the effect was consistently in favour of intervention
Change in psychological symptoms Last follow-up	See comment	See comment	630 (6)	⊕⊕ Low ^10^	The direction of the effect was inconsistent
Change in problems with relationships	See comment	See comment			No RCTs reported this outcome
Change in cognitive function	See comment	See comment			No RCTs reported this outcome
Change in alcohol and other substances misuse Follow-up: 6 months	See comment	See comment			No RCTs reported this outcome
Change in suicidal behaviour	See comment	See comment			No RCTs reported this outcome

CrI: Credible interval; RCT: randomized controlled trial; WL: weight loss. ^1^ We downgraded one level due to imprecision (the number of events was too low to reliably calculate optimal information size), one level due to imprecision (95% CrI includes no effect, and the number of events was too low to reliably calculate optimal information size), and one level for risk of bias (some concern about attrition bias in one study). ^2^ Downgraded one level due to inconsistency (tau = 0.41 [0.04–1.58]), one level due to imprecision (95% CrI includes no effect, and the number of events was too low to reliably calculate optimal information size), and one level for risk of bias (some concern about attrition bias in 4 studies). ^3^ Downgraded one level due to inconsistency (tau = 0.39 [0.04–1.53]), one level due to imprecision (95% CrI includes no effect, and the number of events was too low to reliably calculate optimal information size), and one level for risk of bias (some concern about attrition bias in 4 studies). ^4^ Downgraded one level due to imprecision (95% CrI includes no effect, and the number of events was too low to reliably calculate optimal information size), and one level for risk of bias (some concern about reporting and attrition biases in 2 studies). ^5^ Downgraded one level due to inconsistency (tau = 0.54 [0.04–2.86]), one level due to imprecision (95% CrI includes no effect, and the number of events was too low to reliably calculate optimal information size) and one level for risk of bias (some concern about reporting bias in two studies and attrition bias in 4 studies). ^6^ Downgraded one level due to inconsistency (tau = 0.95 [0.04–5.27]), one level due to imprecision (95% CrI includes no effect, and the number of events was too low to reliably calculate optimal information size), and for risk of bias (some concern about reporting bias in three studies and attrition bias in 4 studies). ^7^ Downgraded two levels due to imprecision (including only one study, small number participants), and one level for risk of bias (some concern about attrition and reporting biases). ^8^ Downgraded one level due to imprecision (95% CrI includes no effect, and the number of events was too low to reliably calculate optimal information size), and one level for risk of bias (some concern about attrition and reporting biases in two studies). ^9^ Downgraded two levels due to imprecision (including only one study, small number participants), and one level for risk of bias (some concern about performance, detection and attrition biases). ^10^ Downgraded one level due to inconsistency and one level for risk of bias (some concern about performance, detection, reporting and attrition biases). Certainty of the evidence expressed in the table by means of ⊕ figures (⊕ very low; ⊕⊕ low).

## Data Availability

The datasets used and/or analysed during the current study are available from the corresponding author on reasonable request.

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
