# Peer review of "Psychological Interventions and Bariatric Surgery among People with Clinically Severe Obesity—A Systematic Review with Bayesian Meta-Analysis"

_nutrients, 2022, doi:10.3390/nu14081592_

Round 1
Reviewer 1 Report
The paper is original and well written. There are too many abbreviations throughout text, maybe they should pooled-up (e.g. I was not able to find the meaning of PPI).
In my opinion some definitions should be better described such as QoL (what kind of assessment in different studies), self efficacy (how was it assessed, again?), etc
Author Response
|
Dear Reviewer,
Thank you very much for your thoughtful comments. We believe they improved the quality of the manuscript.
With best regards,
|
Malgorzata Bala, on behalf of all authors
|
Comment |
Response |
|
The paper is original and well written. There are too many abbreviations throughout text, maybe they should pooled-up (e.g. I was not able to find the meaning of PPI). |
Thank you for pointing this out. We added an abbreviation section as follows: 0. Abbreviations BDI - Beck Depression Inventory score, BF - Bayes factors, BMI - Body Mass Index, BS - bariatric surgery, BT – Behavioural therapy, CAU - care as usual, CBT – Cognitive-behavioural therapy, COI - conflict of interest, CrI - credible intervals, CSO - Clinically severe obesity, GB - Gastric Bypass, HADS - Hospital Anxiety and Depression Scale, IWQOL - Impact of Weight on Quality of Life, LAGB - Laparoscopic adjustable gastric banding, LRYGB Laparoscopic Roux-En-Y Gastric Bypass, LSG - laparoscopic sleeve gastrectomy, NR - not reported, PHQ-9 - Patient Health Questionnaire, POMS - Profile of Mood States, PPI - perioperative psychological interventions, PRISMA - Preferred Reporting Items for Systematic Reviews and Meta-Analyses, QoL - quality of life, RCT - randomized controlled trials, RYGB - Roux-En-Y Gastric Bypass, SEIQoL - Schedule for the Evaluation of Individual Quality of Life, SG - Sleeve Gastrectomy, VGB - Vertical Banded Gastroplasty, WL - weight loss. |
|
In my opinion some definitions should be better described such as QoL (what kind of assessment in different studies), self efficacy (how was it assessed, again?), etc |
Good point. We added the explanations as follows: Because of heterogeneity in outcomes’ presentation, we could not summarize data quantitatively thus only descriptive analysis is presented. Change in self-efficacy was provided in one study. It was measured using General Self Efficacy Scale. The direction of the effect was consistently in favour of intervention in the 6-12 months and the last reported follow-ups and favour of control in the 1-2 years’ follow-up (Appendix J). The certainty of evidence was very low. Change in QoL was provided in three studies. In included trials, several questionnaires were used: 36-Item Short Form Survey - SF-36 (mental and physical components), Schedule for the Evaluation of Individual Quality of Life - SEIQoL, and Impact of Weight on Quality of Life - IWQOL. The direction of the effect was inconsistent (Appendix J). The certainty of evidence was very low for every follow-up.
3.4.2. Secondary Outcomes The trials described different eating behaviours within binge eating or episodes of bulimia. The direction of the effect was inconsistent (Appendix J). The certainty of evidence was very low in case of 6-12 months and 1-2 years follow-ups and low for the last reported follow-up. Two kinds of mood conditions were analysed in included studies: depressive and anxious. Utilized questionnaires included PHQ-9 - Patient Health Questionnaire, HADS - Hospital Anxiety and Depression Scale, BDI - Beck Depression Inventory score, or POMS - Profile of Mood States. The direction of the effect was consistently in favour of intervention at 1-2 years’ follow-up. For the other follow-ups, it was inconsistent (Appendix J). The certainty of evidence was low for all the follow-ups.
|
Reviewer 2 Report
Our experience and several studies suggest that maladaptive eating behaviors such as binge eating, grazing, and a loss of control may impact postsurgical weight outcomes. Several studies showed that eating disorders reactivation or resurgence are the main cause of severe weight regain. This weight regain is observed at least 2 years after the surgery and concern 9/10 patients during the 3-10 years period after the surgery and ranging from 0 to 100% of the weight loss. The major weight regain is correlated to a binge eating reappearance and the loss of control after 2 years. During the early phase, 12-18 first months, weight loss is mainly due to restrictive effect of the surgery and to the homeostatic component of the eating behavior (and other physiologic factors, as genetic, diabetes...). I think that to analyze the effect of psychological intervention during this "honeymoon" period has no interest. Secondly the mean weight loss or the mean loss of excess weight are probably not the best parameter to evaluate. The proportion of patients with a “relapse” defined by abnormal weight regain, specified by a threshold of 10% or 20% could be better as well as the proportion of patients with reactivation of binge eating. The main objective of the multidisciplinary approach is to avoid the eating disorders relapse after the initial period and cannot be evaluated at short term. It is necessary to evaluate these psychological interventions after the weight loss period or the weight nadir.
Pooling different types of procedures is also questionable because the restrictive procedure limits the loss of control whereas the risk of complications such as slipping or esophagus dilation seems enhance in this case. It would be necessary to distinguish each surgical procedure in this analysis.
Minor remark: The meaning of « PPIs » abbreviation has to be defined with the first apparition in the text (Perioperative psychological interventions ?)
Author Response
|
Dear Reviewer,
Thank you very much for your thoughtful comments. We believe they improved the quality of the manuscript.
With best regards,
|
Malgorzata Bala, on behalf of all authors
|
Comment |
Response |
|
Our experience and several studies suggest that maladaptive eating behaviors such as binge eating, grazing, and a loss of control may impact postsurgical weight outcomes. Several studies showed that eating disorders reactivation or resurgence are the main cause of severe weight regain. This weight regain is observed at least 2 years after the surgery and concern 9/10 patients during the 3-10 years period after the surgery and ranging from 0 to 100% of the weight loss. The major weight regain is correlated to a binge eating reappearance and the loss of control after 2 years. During the early phase, 12-18 first months, weight loss is mainly due to restrictive effect of the surgery and to the homeostatic component of the eating behavior (and other physiologic factors, as genetic, diabetes...). I think that to analyze the effect of psychological intervention during this "honeymoon" period has no interest. Secondly the mean weight loss or the mean loss of excess weight are probably not the best parameter to evaluate. The proportion of patients with a “relapse” defined by abnormal weight regain, specified by a threshold of 10% or 20% could be better as well as the proportion of patients with reactivation of binge eating. The main objective of the multidisciplinary approach is to avoid the eating disorders relapse after the initial period and cannot be evaluated at short term. It is necessary to evaluate these psychological interventions after the weight loss period or the weight nadir. |
Thank you for your comment. In our study, we performed subgroup analyses regarding the follow-up investigating the effect in short (6-12 mo), medium (12-24 mo) and the longest observation. We think that we cannot conclude that PPIs do not influence considered outcomes after a short exposition without examining this impact. That is why, we analyse more than one period.
We chose weight parameters as the efficacy parameters, as they are most often used in research (Swierz, Mateusz J., et al. "Systematic review and meta-analysis of perioperative behavioral lifestyle and nutritional interventions in bariatric surgery: a call for better research and reporting." Surgery for Obesity and Related Diseases 16.12 (2020): 2088-2104.).
We agree that you need wait some time for the effects of PPIs, which is why we were not interested in a period shorter than six months.
|
|
Pooling different types of procedures is also questionable because the restrictive procedure limits the loss of control whereas the risk of complications such as slipping or esophagus dilation seems enhance in this case. It would be necessary to distinguish each surgical procedure in this analysis. |
Thank you for this comment. We performed additional subgroup analyses for primary outcomes concerning mixed procedures (RYGB, GB) or both restrictive and mixed procedures (where patients underwent RYGB/GB or VBG/LAGB/SG) (Appendix N), but we did not observe any difference. |
|
Minor remark: The meaning of « PPIs » abbreviation has to be defined with the first apparition in the text (Perioperative psychological interventions ?) |
Thank you for pointing this out. We added the abbreviation of PPI when it first occurred and, additionally, weadded a list of all abbreviations for better readiness as follows: 0. Abbreviations BDI - Beck Depression Inventory score, BF - Bayes factors, BMI - Body Mass Index, BS - bariatric surgery, BT – Behavioural therapy, CAU - care as usual, CBT – Cognitive-behavioural therapy, COI - conflict of interest, CrI - credible intervals, CSO - Clinically severe obesity, GB - Gastric Bypass, HADS - Hospital Anxiety and Depression Scale, IWQOL - Impact of Weight on Quality of Life, LAGB - Laparoscopic adjustable gastric banding, LRYGB Laparoscopic Roux-En-Y Gastric Bypass, LSG - laparoscopic sleeve gastrectomy, NR - not reported, PHQ-9 - Patient Health Questionnaire, POMS - Profile of Mood States, PPI - perioperative psychological interventions, PRISMA - Preferred Reporting Items for Systematic Reviews and Meta-Analyses, QoL - quality of life, RCT - randomized controlled trials, RYGB - Roux-En-Y Gastric Bypass, SEIQoL - Schedule for the Evaluation of Individual Quality of Life, SG - Sleeve Gastrectomy, VGB - Vertical Banded Gastroplasty, WL - weight loss. |
Round 2
Reviewer 2 Report
The only conclusion that we can do is that the study durations are too short to show any effect of the PPI on the bariatric surgery effect. Secondly, the potential interest of the PPI is to be apply in patients with severe eating disorders.
Thank you to re-examine your conclusion to insist that the design of these studies can not permit to show any possible effect
Author Response
Thank you for raising this point. There is some evidence suggesting that a longer interval after bariatric surgery may be associated with weight regain, e.g. Shantavasinkul et al. 2016; however, there are also other studies where weight loss could be observed earlier than two years after BS (Sjostrom et al. 2007, Sugermanet al. 2003). We agree that it would be interesting to see longer term effects of the interventions which we expressed in the implications for research: "High-quality trials with long-term follow-up are required to strengthen the body of evidence as the current evidence is of low to very low methodological quality and at most of anecdotal strength (Bayes factors)". However, we believe our current conclusions are based on the available evidence identified using a rigorous, comprehensive scientific process. Additionally, we believe that our results are not limited to people with severe eating disorders only, because the authors of the studies did not defined their eligibility criteria to include only such populations and no subgroup analyses were possible.